# Five-year outcomes of photodynamic therapy combined with intravitreal injection of ranibizumab or aflibercept for polypoidal choroidal vasculopathy

Kikushima Wataru[1], Atsushi Sugiyama[1], Seigo Yoneyama[1], Mio Matsubara[1], Yoshiko Fukuda[1], Ravi Parikh[2,3], Yoichi Sakurada[1] *

**1** Departments of Ophthalmology, Faculty of Medicine, University of Yamanashi, Kofu, Japan, **2** New York University School of Medicine, New York, New York, United States of America, **3** Manhattan Retina and Eye Consultants, New York, New York, United States of America

* sakurada@yamanashi.ac.jp

**Data Availability Statement:** All relevant data are within the manuscript and its Supporting Information files.

## Abstract

We report 5-year visual and anatomical outcomes after combination therapy of photodynamic therapy (PDT) and intravitreal injection of ranibizumab or aflibercept for polypoidal choroidal vasculopathy (PCV) and predictive factors for visual outcomes at 5-year and time to recurrence. Medical charts were retrospectively reviewed for 43 consecutive eyes with PCV treated with combination therapy of PDT and intravitreal injection of ranibizumab(n = 13) or aflibercept(n = 30) and completed 5-year follow-up. The variants of *ARMS2* A69S and *CFH* I62V were genotyped using TaqMan assay. Best corrected visual acuity (BCVA) significantly improved at 5-year (P = 0.01) with 20% reduction of subfoveal choroidal thickness irrespective of presence or absence of recurrence. Visual improvement was associated with baseline shorter greatest linear dimension (GLD) (P = $1.0 \times 10^{-4}$). Mean time to recurrence was 28.6±23.1 months (95% CI: 21.5–35.7, Median:18.0) and time to recurrence was associated with G allele (protective allele) of *ARMS2* A69S and GLD (P = $4.0 \times 10^{-4}$ and $1.0 \times 10^{-2}$, respectively). Multiple regression analysis revealed that time to recurrence extended by 15.5 months when the G allele of *ARMS2* A69S increased by one allele (TT: 15.7±17.0, TG: 30.8±23.5, GG: 41.1±22.6 months). The combination therapy resulted in a favorable visual outcome for PCV during 5-year follow-up.

## Introduction

Polypoidal choroidal vasculopathy (PCV) often exhibits serosanguinous detachment of retinal pigment epithelium and neurosensory retina and is characterized by aneurysmal dilation with or without branching vascular network on indocyanine green angiography (ICGA).[1, 2]

Several treatment options have been reported to treat PCV including photocoagulation, photodynamic therapy (PDT), intravitreal injection of anti- vascular endothelial growth factor (VEGF) agent and combination therapy of PDT and intravitreal injection of anti-VEGF agent.

**Funding:** JSPS (Japan Society for the promotion of Science) KAKENHI Grant Number 18K16921

**Competing interests:** No authors have competeing interests.

[3–6] Of these treatment options, combination therapy of PDT and intravitreal injection of anti-VEGF agents including ranibizumab and aflibercept is one of the prevalent treatment options along with intravitreal injection of anti-VEGF agent monotherapy. A prospective randomized clinical trial EVERESTII study demonstrated that PDT combined with intravitreal ranibizumab was superior to intravitreal ranibizumab monotherapy in terms of occlusion of polypoidal lesion and visual improvement.[7] Since the follow-up period was short in most studies reporting the combination therapy,[8–12] there has been few reports investigating 5-year outcome of combination therapy of PDT and intravitreal injection of anti-VEGF agent for PCV.[13]

In the present study, we report 5-year visual and anatomical outcomes after combination therapy of PDT and intravitreal injection of ranibizumab or aflibercept for PCV and the predictive factors for visual outcomes at 5 years and time to recurrence during the follow-up.

## Materials and methods

This study was approved for institutional review board of Yamanashi University (Approval No.1957) and was followed in accordance with Declaration of Helsinki. Written informed consent was obtained from each participant. All participants were recruited from the Macular Clinic of the Department of Ophthalmology at University of Yamanashi Hospital between August 2011 and August 2014.

### Treatment and follow-up

Prior to initial treatment, all participants received comprehensive ophthalmic examination including measurement of best-corrected visual acuity(BCVA) using Landolt chart and intraocular pressure, biomicroscopy with or without a 78 diopter lens, color fundus photography, fluorescein and indocyanine green angiography(FA/ICGA) (HRA-2, Heidelberg Engineering, Dossenheim, Germany), spectral-domain optical coherence tomography(SD-OCT) (Spectralis ver5.4 HRA+OCT). All SD-OCT images were obtained by both horizontal and vertical lines through the fovea using EDI-mode. Central retinal thickness was defined as a vertical distance from bottom of RPE to inner limiting membrane at the fovea and subfoveal choroidal thickness was defined as a vertical distance from bottom of RPE to choroidoscleral border at the fovea as measured by SD-OCT.

Diagnosis of PCV was made as previously described.[14] All PCV cases exhibited solitary or clusters of polypoidal dilation with or without abnormal vascular networks in the superficial choroid on ICGA and irregular RPE elevations with serous and/or hemorrhagic detachment of neurosensory retina or RPE on SD-OCT.

Exclusion criteria was the case with massive subretinal hemorrhage with or without vitreous hemorrhage, dense cataract, history of recent thromboembolic events, or pregnancy.

All participants received intravitreal injection of ranibizumab (0.05mg/0.05ml) or aflibercept (0.2mg/0.05ml) 1 week before PDT (1 injection and 1PDT). The initial treatments were chosen according to time period. From August 2011 to 2012 December, combination therapy of PDT and intravitreal ranibizumab injection was administrated and from January 2013 to August 2014 combination therapy of PDT and intravitreal aflibercept was administrated. Standard PDT (a laser light at 689 nm delivered at a dose of $50J/cm^2$ at an intensity of $600mW/cm^2$ over 83 sec with verteporfin $6mg/m^2$) was conducted. Greatest linear dimension (GLD) was determined as maximum length to cover the polypoidal lesion and branching vascular networks. Spot size was defined as the length of 1000μm added to GLD.

Follow-up examination included assessment of BCVA using Landolt chart, intraocular pressure, biomicroscopy with or without a 76 D lens, and SD-OCT, and was performed every

3 months until recurrent exudation developed. Recurrence was defined as newly developed hemorrhage on fundoscopy or subretinal fluid detected by SD-OCT. Additional FA/ICGA was performed when recurrent exudation was seen. When ICGA showed residual or recurrent polypoidal lesion, additional combination therapy (1injection and 1 PDT) was administrated in the same fashion as the initial combination therapy. When ICGA exhibited abnormal vascular network without polyp, additional intravitreal injection of anti-VEGF agent was administrated. After first recurrence, patients were followed every month and PRN treatment was performed until exudation including subretinal fluid and intraretinal fluid was completely absorbed.

## Genotyping

Genomic DNA was obtained from peripheral blood using PURE GENE Isolation Kit (Gentra Systems, Minneapolis, US). The genotyping of *ARMS2* A69S (rs10490924) and *CFH* I62V (rs800292) was performed for all participants using TaqMan assays on ABI7300/7500 Real Time PCR System (Applied Biosystems, Foster City, US) as we recently described.[15]

## Statistical analysis

Statistical analysis was conducted using DR. SPSS (IBM, Tokyo, Japan). A chi-square test was used to evaluate differences of categorical variables between 2 groups. Mann-Whitney U test was used to evaluate differences of continuous variables between 2 groups. Decimal BCVA was converted to log MAR BCVA for statistical analysis. A paired t-test was used to compare log MAR BCVA before and after treatment. Cox-regression analysis was performed to investigate risk factors for recurrence and multiple regression analysis was performed to investigate the factors associated with time to recurrence. A p-value less than 0.05 was considered a statistical significance.

## Results

During the study period, 43 eyes from 43 patients were included in the present study. Table 1 shows the baseline characteristics in 43 subjects. Table 2 shows the baseline characteristics comparison of 2 treatment groups. There were no significant differences between 2 treatment groups but central retinal thickness, in which IVR+PDT group had greater CRT than IVA +PDT group at baseline. Fig 1 shows changes of BCVA (Fig 1A) and changes of BCVA gains (Fig 1B) following combination therapy. After combination therapy, BCVA significantly improved (p<0.01) compared with baseline value throughout 5-year follow-up. Mean logMAR BCVA improved from 0.55±0.28 at baseline to 0.40±0.40 at 60 month. Mean logMAR BCVA gains at 2-year from baseline were greatest throughout 5-year follow-up. Compared with BCVA at 2-year, those values at 4-year and 5-year were significantly worse (p = 0.005 and p = 0.001, respectively). Fig 2 shows BCVA gains from baseline at 5-year in each treatment group. There were no significant differences in BCVA gains between the 2 treatment groups throughout 5-year follow-up. Factors at 60 months multivariate linear regression analysis revealed that shorter GLD (p = $1.0 \times 10^{-4}$) were associated with better BCVA at 60 months while other factors such as age, type of anti-VEGF agent, and central retinal thickness were not associated with improved BCVA (Table 3). Another multivariate linear regression analysis associated with the BCVA gains from baseline to 5-year revealed that shorter GLD (p = $1.0 \times 10^{-4}$) and worse baseline BCVA(p = $7.0 \times 10^{-4}$) was associated with greater BCVA gains (Table 4). T allele (risk allele) of *ARMS2* A69S was the only variable associated with recurrence of exudation from PCV (P = 0.002, Table 5). Fig 3(A), 3(B) and 3(C) demonstrates recurrence-free proportion according to *ARMS2* A69S genotypes(A), *CFH* I62V genotypes(B),

**Table 1. Clinical and genetic characteristics of the subjects.**

| | |
|---|---|
| Age | 72.8±7.6(57–92) |
| Male gender | 30(69.8%) |
| Baseline log MAR BCVA | 0.55±0.27(0.10–1.22) |
| Greatest linear dimension (μm) | 1666±797(450–3600) |
| Central Retinal Thickness (μm) | 368.1±99.9(178–610) |
| Subfoveal Choroidal Thickness (μm) | 257.0±87.7(95–480) |
| Initial Anti-VEGF agent | |
| Ranibizumab | 13(30.2%) |
| Aflibercept | 30(69.8%) |
| Number of Polyps | 1.79±1.15(1.0)1-5 |
| *ARMS2* A69S(rs10490924) | Genotype frequency |
| TT | 13(30.2%) |
| TG | 20(46.5%) |
| GG | 10(23.3%) |
| *CFH* I62V(rs800292) | Genotype frequency |
| GG | 25(58.1%) |
| GA | 16(37.2%) |
| AA | 2(4.7%) |
| Lens Status (phakic eye) | 38(88.4%) |
| Current Smoker | 5(11.6%) |

ARMS: age-related maculopathy susceptibility, BCVA: best-corrected visual acuity

CFH: complement factor H, GLD: greatest linear dimension, log MAR: logarithm of the minimal angle resolution,

BCVA: best-corrected visual acuity, VEGF: vascular endothelial growth factor

**Table 2. Baseline characteristics comparison of 2 treatment groups.**

| | IVR+PDT(n = 13) | IVA+PDT(n = 30) | P value |
|---|---|---|---|
| Age | 71.0±5.3(62–80) | 73.6±8.3(57–92) | 0.23 |
| Male gender | 10(76.9%) | 20(66.7%) | 0.50 |
| Baseline log MAR BCVA (range) | 0.56±0.25(0.22–1.00) | 0.55±0.29(0.10–1.22) | 0.72 |
| Greatest linear dimension (μm)(range) | 1492±881(600–3600) | 1742±761(450–3500) | 0.20 |
| Central retinal thickness (μm) (range) | 422±97(283–610) | 345±93(178–588) | 0.01 |
| Subfoveal choroidal thickness (μm)(range) | 275±96(145–480) | 249±85(95–419) | 0.48 |
| Number of polyps(range) | 2.00±1.22 (1–5) | 1.70±1.12 (1–4) | 0.32 |
| *ARMS2* A69S(rs10490924) | | | 0.32 |
| TT | 4(30.8%) | 9(30%) | |
| TG | 8(61.6%) | 12(40%) | |
| GG | 1(7.6%) | 9(30%) | |
| *CFH* I62V(rs800292) | | | 0.10 |
| GG | 5(38.5%) | 20(66.7%) | |
| GA | 7(53.8%) | 9(30%) | |
| AA | 1(7.7%) | 1(3.3%) | |
| Current Smoker | 2(15.4%) | 3(10%) | 0.61 |
| Lens Status (phakic eye) | 12(92.3%) | 26(86.7%) | 0.60 |

ARMS: age-related maculopathy susceptibility, BCVA: best-corrected visual acuity

CFH: complement factor H, GLD: greatest linear dimension, log MAR: logarithm of the minimal angle resolution,

BCVA: best-corrected visual acuity, VEGF: vascular endothelial growth factor

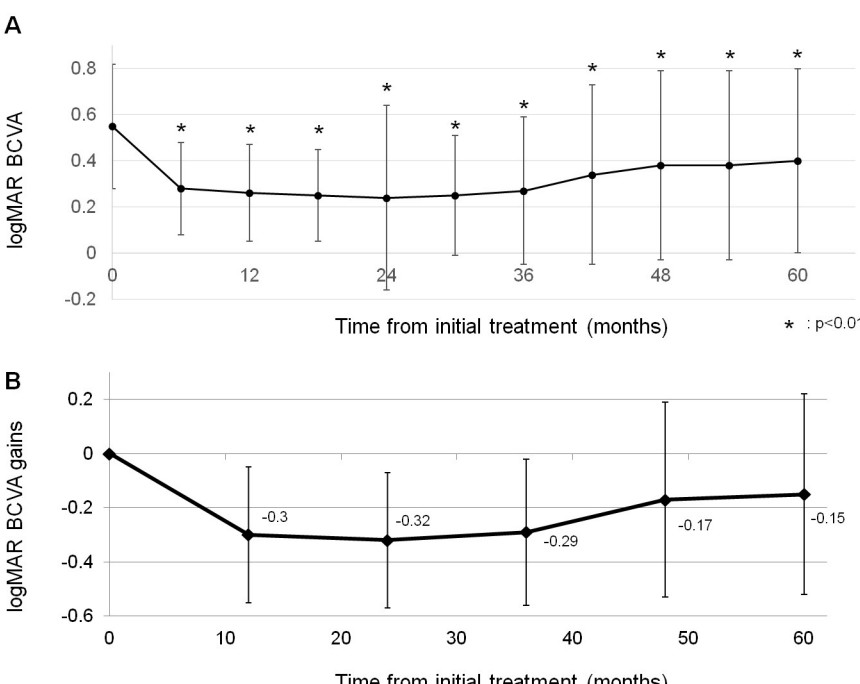

**Fig 1.** (A) Changes of best-corrected visual acuity after the combination therapy. (B) Changes of best-corrected visual acuity gains after the combination therapy.

and anti-VEGF agents(C). *ARMS2* A69S genotype TT demonstrated a lower proportion of recurrence-free eyes compared with genotypes TG and GG (p = 0.019). Across *CFH* genotypes and anti-VEGF agents no statistically significant difference in recurrence free proportions were found (p = 0.95 and p = 0.48).

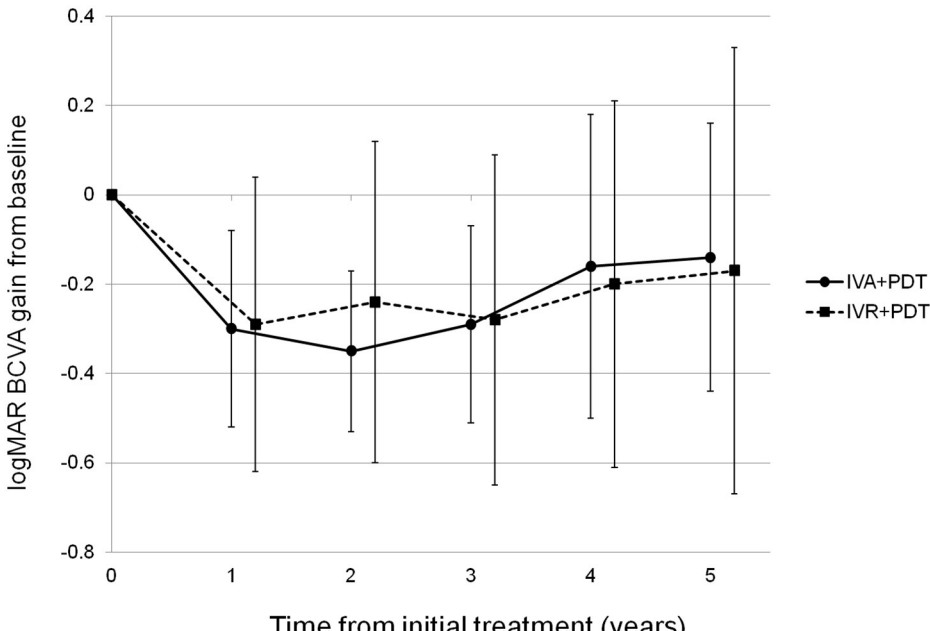

**Fig 2. Best-corrected visual acuity gain after the combination therapy according to anti-VEGF agents.**

**Table 3. Results of multivariate linear regression analyses associated with the logarithm of the minimal angle resolution best-corrected visual acuity at 60 months.**

| Variables | β-coefficient | p-value |
|---|---|---|
| Age | 0.004 | 0.58 |
| Sex | 0.23 | 0.05 |
| Baseline BCVA (log MAR) | 0.33 | 0.07 |
| Greatest linear dimension (μm) | $2.8 \times 10^{-4}$ | $1.0 \times 10^{-4}$ |
| Central retinal thickness (μm) | -0.001 | 0.07 |
| Subfoveal choroidal thickness (μm) | $-8.0 \times 10^{-4}$ | 0.15 |
| Initial anti-VEGF agent | -0.12 | 0.27 |
| (1: aflibercept, 0: ranibizumab) | | |
| Current smoker | 0.13 | 0.39 |
| ARMS2 A69S T-allele | 0.11 | 0.09 |
| CFH I62V G-allele | 0.06 | 0.51 |
| Number of Polyps | -0.02 | 0.61 |
| Lens status (0: phakic, 1: pseudophakic) | 0.03 | 0.83 |

BCVA: best corrected visual acuity

Log MAR: logarithm of the minimal angle resolution

VEGF: vascular endothelial growth factor

ARMS: age-related maculopathy susceptibility

CFH: complement factor H

1. Recurrence-free proportion after the initial combination therapy among *ARMS2* A69S genotypes.
   Mean time to recurrence was 28.6 months (95%CI: 21.5–35.7, median:18 months) in all subjects. Time to recurrence was significantly different among *ARMS2* A69S genotypes (P = 0.019, log-rank test). Mean time to recurrence was 41.1±22.6 months in GG genotype (95%CI:24.9–57.3, median:54 months), 30.8±23.5 months in TG genotype

**Table 4. Results of multivariate linear regression analyses associated with the logarithm of the minimal angle resolution best-corrected visual acuity gains from baseline to at 60 months.**

| Variables | β-coefficient | p-value |
|---|---|---|
| Age | 0.004 | 0.58 |
| Sex | 0.23 | 0.05 |
| Baseline BCVA (log MAR) | -0.67 | $7.0 \times 10^{-4}$ |
| Greatest linear dimension (μm) | $2.8 \times 10^{-4}$ | $1.0 \times 10^{-4}$ |
| Central retinal thickness (μm) | -0.001 | 0.066 |
| Subfoveal choroidal thickness (μm) | $-8.0 \times 10^{-4}$ | 0.15 |
| Initial anti-VEGF agent | -0.12 | 0.27 |
| (1: aflibercept, 0: ranibizumab) | | |
| Current smoker | 0.13 | 0.39 |
| ARMS2 A69S T-allele | 0.11 | 0.09 |
| CFH I62V G-allele | 0.05 | 0.51 |
| Number of Polyps | -0.02 | 0.61 |
| Lens status (0: phakic, 1: pseudophakic) | 0.03 | 0.83 |

BCVA: best corrected visual acuity

Log MAR: logarithm of the minimal angle resolution

**Table 5. Cox regression analysis of factors associated with recurrence.**

| Variables | p-value | Hazard ratio | 95% confidence interval |
|---|---|---|---|
| Age | 0.23 | 1.05 | 0.97–1.13 |
| Male gender | 0.14 | 2.23 | 0.77–6.49 |
| Baseline logMAR BCVA | 0.68 | 0.68 | 0.11–4.34 |
| Greatest linear dimension | 0.13 | 1.00 | 1.00–1.00 |
| Central retinal thickness | 0.25 | 1.00 | 0.99–1.00 |
| Subfoveal choroidal thickness | 0.58 | 1.00 | 0.99–1.00 |
| Aflibercept | 0.23 | 0.53 | 0.19–1.50 |
| Current smoker | 0.66 | 0.71 | 0.15–3.26 |
| ARMS2 A69S T allele | 6.0×10−4 | 3.38 | 1.68–6.79 |
| CFH I62V G allele | 0.57 | 1.24 | 0.59–2.63 |
| Number of Polyps | 0.13 | 1.34 | 0.91–1.98 |
| Lens status(0:phakic, 1:pseudophakic) | 0.07 | 0.21 | 0.04–1.16 |

ARMS2: age-related maculopathy susceptibility, BCVA: best-corrected visual acuity, CFH: complement factor H, logMAR: logarithm of the minimal angle resolution

(95%CI:19.7–41.8, median:24 months), 15.7±17.0 months in TT genotype (95%CI:5.4–26.0, median:9 months).

2. Recurrence-free proportion after the initial combination therapy among *CFH* I62V genotypes.
   Time to recurrence was not significantly different among *CFH* I62V genotypes (P = 0.95, log-rank test). Mean time to recurrence was 31.5±40.3 months in AA genotype

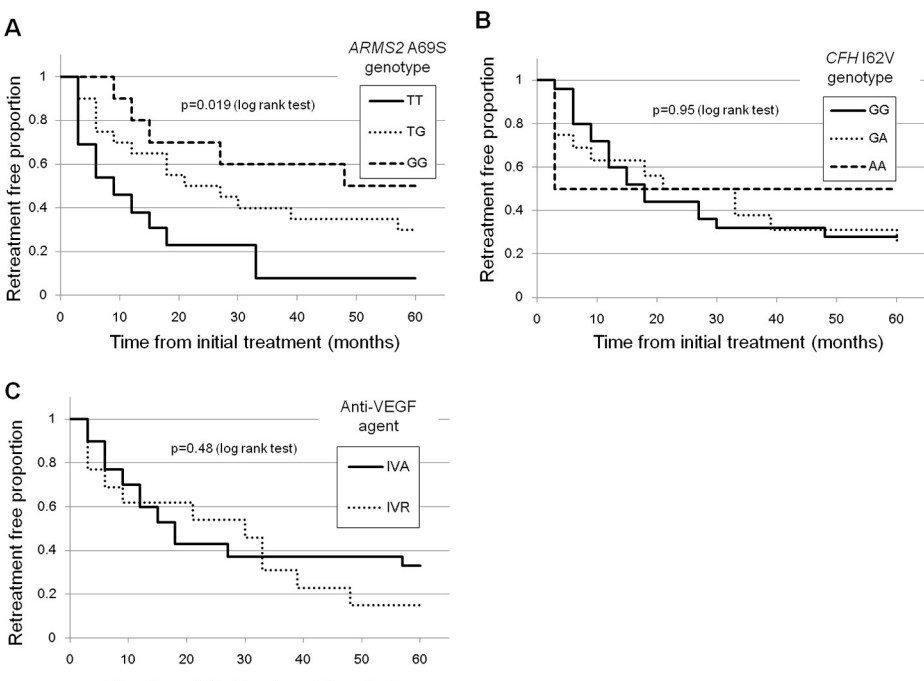

**Fig 3. Kaplan-Meier survival curve associated with recurrence-free proportion after the initial combination therapy.**

**Table 6. Results of multivariate linear regression analyses associated with time to recurrence (backward stepwise selection method).**

| Variables | β-coefficient | p-value |
|---|---|---|
|  | 39.2 |  |
| Sex | -13.2 | 0.053 |
| ARMS2 A69S G-allele | 15.5 | $4.0 \times 10^{-4}$ |
| Greatest linear dimension (μm) | -0.01 | 0.01 |

ARMS: age-related maculopathy susceptibility

Age, sex, smoking status, baseline best-corrected visual acuity, baseline central retinal thickness, baseline subfoveal choroidal thickness, initial anti-VEGF agent, and CFH I62V were eliminated in this analysis.

(95%CI:3–60,median:30 months), 29.3±23.9 months in GA genotype (95%CI:16.5–42.0, median:27 months), 28.0±22.5 months in GG genotype (95%CI:18.7–37.3, median:18 months).

3. Recurrence-free proportion after the initial combination therapy depending on anti-VEGF agents.
   Time to recurrence was not significantly different depending on anti-VEGF agents
   (P = 0.48, log-rank test). Mean time to recurrence was 29.4±24.2 months in IVA group (95%CI:20.4–38.4, median:18 months), 26.8±21.2 months (95%CI:14.0–39.6, median:30 months) in IVR group.

GG genotypes at A69S of *ARMS2* recurred less frequently compared with other genotypes. Multivariate linear regression analyses (backward stepwise selection method, Table 6) revealed that increased time to recurrence was associated with G allele (protective allele) of *ARMS2* A69S and shorter GLD and time to recurrence was estimated as follows: 39.2+15.5× (number of G allele)-0.01× GLD (μm) (months). Thus, for each G allele the recurrence free time increased by 15.5 months on average. Annual number of additional treatments including anti-VEGF injection and the combination therapy was shown in Table 7. Mean number of additional injections and the combination therapy during 5-year follow-up was 7.5 and 0.5, respectively. Table 8 shows changes of subfoveal choroidal thickness depending on presence or

**Table 7. Annual number of additional injection/combination therapy in all patients.**

| Months after initial treatment | 0-12M | 13-24M | 25-36M | 37-48M | 49-60M | Total (0-60M) |
|---|---|---|---|---|---|---|
| mean(median) number of additional injections | 0.47±0.93 (0.0) | 1.09±1.60 (0.0) | 1.84±2.34 (1.0) | 1.93±2.09 (1.0) | 2.19±2.22 (2.0) | 7.51±7.25 (7.0) |
| mean(median) number of additional combination therapies | 0.12±0.32 (0.0) | 0.12±0.32 (0.0) | 0.16±0.43 (0.0) | 0.09±0.29 (0.0) | 0.02±0.15 (0.0) | 0.51±1.01 (0.0) |

**Table 8. Changes of subfoveal choroidal thickness in patients with or without recurrence.**

| Months after initial treatment | 0M (baseline) | 12M | 24M | 36M | 48M | 60M |
|---|---|---|---|---|---|---|
| SCT (n = 43, all patient) | 257.0±87.7 | 222.5±87.4 | 218.1±81.9 | 218.1±98.9 | 210.6±95.0 | 206.3±92.7 |
| P value (vs baseline) |  | $7.1 \times 10^{-5}$ | $3.2 \times 10^{-5}$ | $4.0 \times 10^{-5}$ | $1.6 \times 10^{-5}$ | $1.7 \times 10^{-6}$ |
| SCT (n = 31, recurrence group) | 256.3±75.3 | 225.2±80.3 | 220.6±73.3 | 220.6±93.4 | 210.0±90.3 | 207.3±88.7 |
| P value (vs baseline) |  | $3.6 \times 10^{-3}$ | $2.2 \times 10^{-3}$ | $3.1 \times 10^{-3}$ | $8.0 \times 10^{-4}$ | $2.0 \times 10^{-4}$ |
| SCT (n = 12, non-recurrence group) | 258.9±117.8 | 215.6±107.4 | 211.7±104.3 | 211.5±116.1 | 212.2±110.6 | 203.7±106.5 |
| P value (vs baseline) |  | $4.3 \times 10^{-3}$ | $2.4 \times 10^{-3}$ | $9.0 \times 10^{-4}$ | $2.5 \times 10^{-3}$ | $1.5 \times 10^{-3}$ |

SCT: subfoveal choroidal thickness

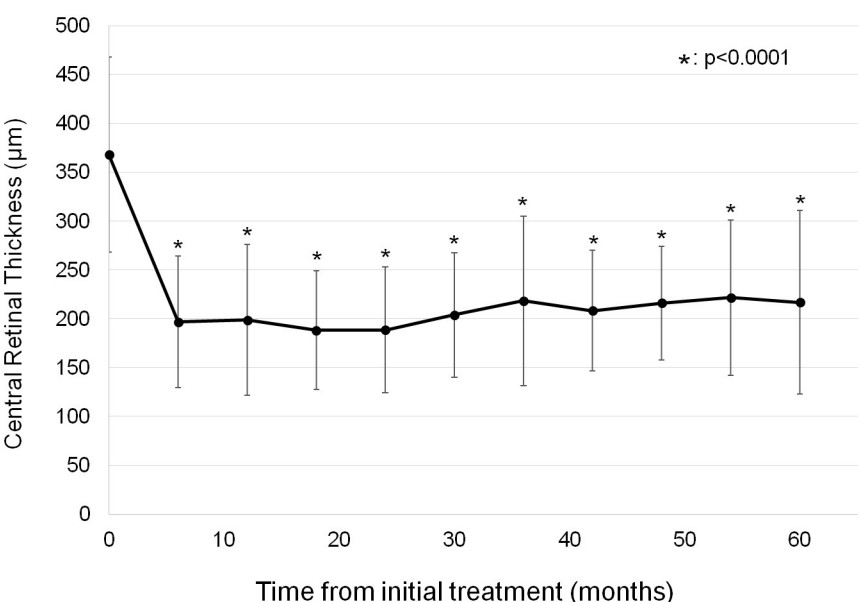

**Fig 4. Changes of central macular thickness after the initial combination therapy.**

absence of recurrence. Subfoveal choroidal thickness significantly decreased throughout the follow-up period. Irrespective of presence or absence of recurrence, subfoveal choroidal thickness decreased by approximately 20% at 60 months. Fig 4 shows changes of central macular thickness after the initial combination therapy. Compared with baseline value, central macular thickness significantly decreased (p = <0.001) throughout the follow-up period and mean CRT decreased from 368±100 μm at baseline to 217±94 μm at 60 month.

Compared with baseline value (368±100),central macular thickness significantly decreased to 199±77 (P = $2.5 \times 10^{-12}$), 189±64 (P = $1.2 \times 10^{-14}$), 218±87 (P = $2.8 \times 10^{-11}$), 216±85 (P = $3.9 \times 10^{-15}$), and 217±94 (P = $2.7 \times 10^{-11}$) at 12-month, 24-month, 36-month, 48-month, and 60-month, respectively.

## Discussion

We investigated long-term visual and anatomical outcomes after the combination therapy involving PDT and intravitreal injection of ranibizumab or aflibercept for PCV. In the present study, significant visual gain was obtained at 1 year and was maintained through 5-years. Several studies reported long-term results of the combination therapy involving PDT and anti-VEGF agents for PCV.[13, 16, 17] Miyata et al. reported 5-year results after combination therapy involving PDT and intravitreal injection of ranibizumab for 20 eyes with PCV. In their report, BCVA significantly improved at 1 year, but thereafter BCVA declined and returned to baseline value at 5 year. Similarly, in the present study, there has been the trend of BCVA deterioration after 3 years from initial treatment, but the significant BCVA improvement was maintained through 5 years. One possible reason for the differences of visual outcome between the 2 studies is that, in this study, mean age of the patients was younger than that in the Miyata's report. We reported that in the previous study, older age was associated with recurrence of PCV lesions after the combination therapy.[18] The difference of mean age between the 2 studies might affect the visual outcome. Differences in baseline characteristic between the 2 studies might result in different results. To the best of our knowledge, this is the first report demonstrating long-term results of combination therapy involving IVA and PDT for PCV.

We compared the long-term visual outcome between IVA+PDT group and IVR+PDT group; however, there were no significant differences in visual outcomes between the 2 groups. In this study, predictive factors of visual gain at 5 years was shorter GLD. Regarding the association between visual improvement and shorter GLD, Tsujikawa et al.[19] demonstrated that eyes with a smaller lesion size (less than one-disc diameter) determined by ICGA have a better visual prognosis than eyes with larger lesion size (larger than one-disc diameter) over 24-month follow up. In addition to the report, several studies showed that GLD is one of the determinant factors for BCVA after treatment for PCV.[20–22]

Regarding morphological changes during the follow-up period, central retinal thickness and subfoveal choroidal thickness significantly decreased at 1 year and maintained throughout 5-year. Several studies demonstrated that subfoveal choroidal thickness reduced after PDT in eyes with PCV.[22, 23] Maruko et al.[23] reported that subfoveal decreased choroidal thickness increased and returned to the baseline level when recurrent exudation occurred in eye with PCV. On the basis of the previous report, we subdivided eyes with PCV into recurrent group or non-recurrent group during 5-year follow up. Subfoveal choroidal thickness decreased by approximately 20% from baseline to 5-year in both groups. However, there was not a significant difference of subfoveal choroidal thickness changes between the 2 groups.

We genotyped two major genetic variants susceptible to PCV including *ARMS2* A69S and *CFH* I62V. Especially, variants of *ARMS2* A69S have been reported to be associated with various clinical phenotype including lesion size, subfoveal choroidal thickness and bilateral involvement.[24–26] In addition to the association of PCV phenotypes, several investigators reported the relationship between *ARMS2* A69S and treatment response for PCV including PDT monotherapy, intravitreal anti-VEGF treatment monotherapy and combination therapy. [18, 22, 27–31] Most studies focused on visual outcomes or recurrence after treatment and demonstrated that at-risk allele homozygosity (TT genotype) at A69S of *ARMS2* were more likely to recur compared with other genotypes. In addition to revealing that recurrence was more frequently seen in risk allele of *ARMS2* A69S dependent fashion, multiple regression analysis revealed that time to recurrence extended by 15.5 months when G allele (non-risk allele) of *ARMS2* A69S increased by one allele. To the best of our knowledge, this is the first report revealing that non-risk variants of the *ARMS2* gene are associated with time to recurrence as well as chance of recurrence. These finding might be helpful for both physicians and patients when considering an optimal follow-up duration.

In the present study, intravitreal injection of anti-VEGF agents was administrated a week before PDT. The reason why we chose this method was that, PDT causes photochemical thrombotic occlusion of polypoidal lesions, while also induces the upregulation of VEGF as an adverse side effect. Initial injection of anti-VEGF agents is expected to decrease the high intra-ocular concentration of VEGF, which causes exudation of both intraretinal and subretinal fluid.

There are several limitations in the current study. Major limitation of this study is a relatively small sample size and a retrospective nature of analysis. Secondly, ICGA was performed only when recurrent exudation developed though all patients received ICGA prior to the initial treatment. Therefore, we cannot discuss the progression or regression of branching vascular network and polypoidal lesion during the follow-up period. Thirdly, early ICGA images were not performed for all patients, therefore we could not judge presence or absence of feeder vessels of polypoidal lesions. For this reason, we did not distinguish the PCV subtypes. A large-scale prospective study would be needed to confirm the present tentative conclusion.

In summary, the combination therapy of PDT and intravitreal injection of ranibizumab or aflibercept is an effective treatment for PCV over 5-year follow-up. Time to recurrence is

associated with G allele of *ARMS2* A69S and GLD. When G allele increases by one allele, it is estimated that time to recurrence extends by 15.5 months.

## Supporting information

**S1 Data.**
(XLSX)

## Author Contributions

**Conceptualization:** Kikushima Wataru, Mio Matsubara, Yoshiko Fukuda.

**Data curation:** Kikushima Wataru, Atsushi Sugiyama, Seigo Yoneyama.

**Formal analysis:** Kikushima Wataru.

**Funding acquisition:** Kikushima Wataru, Mio Matsubara.

**Methodology:** Atsushi Sugiyama, Yoshiko Fukuda.

**Resources:** Atsushi Sugiyama, Mio Matsubara.

**Supervision:** Yoshiko Fukuda, Ravi Parikh, Yoichi Sakurada.

**Writing – original draft:** Kikushima Wataru, Yoshiko Fukuda.

**Writing – review & editing:** Atsushi Sugiyama, Ravi Parikh, Yoichi Sakurada.

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
