## [Decision Letter · Decision Letter 0]

31 Dec 2019

PONE-D-19-34381

Five-year outcomes of photodynamic therapy combined with intravitreal injection of ranibizumab or aflibercept for polypoidal choroidal vasculopathy

PLOS ONE

Dear Dr Sakurada,

Thank you for submitting your manuscript to PLOS ONE. After careful consideration, we feel that it has merit but does not fully meet PLOS ONE’s publication criteria as it currently stands. Therefore, we invite you to submit a revised version of the manuscript that addresses the points raised during the review process.

The reviewers have requested the addition of more information and new tables and figures to make the data more complete. Moreover the literature review requires to be updated and more complete.

We would appreciate receiving your revised manuscript by Feb 14 2020 11:59PM. To enhance the reproducibility of your results, we recommend that if applicable you deposit your laboratory protocols in protocols.io, where a protocol can be assigned its own identifier (DOI) such that it can be cited independently in the future. For instructions see: http://journals.plos.org/plosone/s/submission-guidelines#loc-laboratory-protocols

We look forward to receiving your revised manuscript.

Kind regards,

Michael R Hamblin

Academic Editor

PLOS ONE

Journal Requirements:

Reviewers' comments:

Reviewer's Responses to Questions

**Comments to the Author**

1. Is the manuscript technically sound, and do the data support the conclusions?

Reviewer #1: Yes

Reviewer #2: Yes

2. Has the statistical analysis been performed appropriately and rigorously? 

Reviewer #1: Yes

Reviewer #2: Yes

3. Have the authors made all data underlying the findings in their manuscript fully available?

Reviewer #1: Yes

Reviewer #2: No

4. Is the manuscript presented in an intelligible fashion and written in standard English?

Reviewer #1: Yes

Reviewer #2: Yes

5. Review Comments to the Author

Reviewer #1: The article is well written, and the first article regarding long-term results of combination of PDT and IVA for PCV. The article can accepted after major revision.

Comments:

1. A table should be added regarding Baseline characteristics comparison of 2 combination therapy. Are they matched in age, gender, etc..?

2. A table should be added about “multivariate linear regression analyses associated with the logarithm of the minimal angle resolution best-corrected visual acuity gains from baseline to at 60 months”

3. Lens status and polyp number should be added as an analyzed factor (phakic or pseudophakic) in either baseline characteristics, final VA, and VA gains

4. Each combination therapy included one PDT with one injection or two injections? After recurrence, how many injections were given? One injection or 3 monthly injections? If SRF or IRC persisted after injection or combination therapy , how did you do?

5. It seems BCVA at year 1, 2, and 3 were better than that at year 4 and 5. Can you compare the difference of BCVA at each year? Maybe there is still the trend of VA deterioration after long-term follow-up, which similar to the prior results of the study.

6. BCVA of PDT+IVA and PDT+IVR should be drawn in another figure and compared each year results. It will be an interesting result because this was the first long-term report about PDT+IVA for PCV. The authors should emphasize this unique point.

7. Are there any cases presenting with massive subretinal hemorrhage with or without vitreous hemorrhage? Do you still use combination therapy at the first place, instead of use of intravitreal tPA or gas? Or you exclude this kind of cases? If so, you should add exclusion criteria.

8. If the case with dense cataract and PCV as the first presentation, how did you do? Or you exclude this kind of cases?

9. When you use intravitreal anti-VEGF, do you exclude the cases with recent thromboembolic events or pregnancy?

10. Table 1 did not have the range of Age, Baseline log MAR BCVA, Greatest linear dimension (μm), Central Retinal Thickness (μm), and Subfoveal Choroidal Thickness

11. Can you list in a table or describe in the article about final log MAR BCVA, Greatest linear dimension (μm), Central Retinal Thickness (μm), and Subfoveal Choroidal Thickness

Reviewer #2: It is useful to publish the data presented as the long term (5 year) outcomes are valuable information. The combination of anti-VEGF + PDT is at present a treatment of choice for PCV. I have the following remarks:

1. Line 44: Please leave out the statement starting with "To date PCV has been considered ...". This is not generally true.

2. In the description of the PDT important parameters are left out. Over what time was the injection performed? How long was the delay between injection and light application?

3. At the bottom of page 5: Please describe more precisely how the follow-up treatments were done. When was only anti-VEGF applied, at what frequency, and what conditions? When combination therapy was applied in the follow-up, did anti-VEGF precede PDT, etc.

4. There are very important differences in the outcome when compared with the data of reference 13 (Miyata et. al.). This needs to be discussed in some detail.

5. The literature reports are rather incomplete. The literature reported is to a significant extent self-citation. See for instance P. Jain et. al. Indian J. Ophthalmol. 2018, Jingyuan Yang et. al. BMC Ophthalmology December 2019, and Qian T. et. al. Eur. J. Clin. Invest. 2018. among others.

6. Discussion of the influence of the ARMS2 gene as done by the present authors is of interest. Why not also discuss the results in terms of the subtypes PCV1 AND PCV2.

7. Other authors have done this combination therapy by treating first with PDT and the following up with anti-VEGF. Please discuss your choice, and why you think it is preferable.

6. PLOS authors have the option to publish the peer review history of their article (what does this mean?). If published, this will include your full peer review and any attached files.

Reviewer #1: No

Reviewer #2: No

---

## [Author Response · Author response to Decision Letter 0]

28 Jan 2020

Dear Editor:

We thank you, and the reviewer for timely and constructive feedback on our manuscript. We appreciate the opportunity to respond.

Sincerely

Yoichi Sakurada, M.D., Ph.D

Reviewer #1: The article is well written, and the first article regarding long-term results of combination of PDT and IVA for PCV. The article can be accepted after major revision.

Reply: We thank you for your constructive feedback and insightful comments.

Comments:

1. A table should be added regarding Baseline characteristics comparison of 2 combination therapy. Are they matched in age, gender, etc.?

Reply: Based on the suggestion, we added a new table as Table 2, and we added the explanation of the table in the Results section as follows " Table 2 shows the baseline characteristics comparison of 2 treatment groups. There were no significant differences between 2 treatment groups but central reinal thickness, in which IVR+PDT group had greater CRT than IVA+PDT group at baseline."

2. A table should be added about “multivariate linear regression analyses associated with the logarithm of the minimal angle resolution best-corrected visual acuity gains from baseline to at 60 months”

Reply: On the basis of the suggestion above, we added a new table as Table 4, and we added the explanation of the table in the Results section as follows "Another multivariate linear regression analysis associated with the BCVA gains from baseline to 5-year revealed that shorter GLD(p=1.0×10-4) and worse baseline BCVA(7.0×10-4) was associated with greater BCVA gains(Table 4)."

3. Lens status and polyp number should be added as an analyzed factor (phakic or pseudophakic) in either baseline characteristics, final VA, and VA gains

Reply: Thank you for the advice. We added lens status and number of polyps as additional factors to the baseline characteristics and described the results in Table 1, Table2, Table 3 and Table 4 in the Results Section.

4. Each combination therapy included one PDT with one injection or two injections? After recurrence, how many injections were given? One injection or 3 monthly injections? If SRF or IRC persisted after injection or combination therapy, how did you do?

Reply: We added the detailed explanation regarding the retreatment in the Materials and Methods section as follows "Recurrence was defined as newly developed hemorrhage on fundoscopy or subretinal fluid detected by SD-OCT. Additional FA/ICGA was performed when recurrent exudation was seen. When ICGA showed residual or recurrent polypoidal lesion, additional combination therapy (1 injection and 1 PDT) was administrated in the same fashion as the initial combination therapy. When ICGA exhibited abnormal vascular network without polyp, additional intravitreal injection of anti-VEGF agent was administrated. After first recurrence, patients were followed every month and PRN treatment was performed until exudation including subretinal fluid and intraretinal fluid was completely disappeared."

5. It seems BCVA at year 1, 2, and 3 were better than that at year 4 and 5. Can you compare the difference of BCVA at each year? Maybe there is still the trend of VA deterioration after long-term follow-up, which similar to the prior results of the study.

Reply: Thank you for the advice. We added a new figure showing the BCVA gains in each year as Fig 1B and described the result in the Results section as follows " Mean log MAR BCVA improved from 0.55±0.28 at baseline to 0.40±0.40 at 60-month. Mean log MAR BCVA gains at 2-year from baseline were greatest thoughout 5-year follow-up. Compared with BCVA at 2-year, those values at 4-year and 5-year were significantly worse (p=0.005 and p=0.001, respectively)."

6. BCVA of PDT+IVA and PDT+IVR should be drawn in another figure and compared each year results. It will be an interesting result because this was the first long-term report about PDT+IVA for PCV. The authors should emphasize this unique point.

Reply: Thank you for the positive comments. On the basis of your suggestion, we added a new figure as Fig 2 and described the result in the Results section as follows "Fig 2 shows BCVA gains from baseline at 5 years in each treatment group. There were no significant differences in BCVA gains between the 2 treatment groups throughout 5-year follow-up." We also added the new sentences in the Discussion section as follows " To the best of our knowledge, this is the first report demonstrating long-term results of combination therapy involving IVA and PDT for PCV. We compared the long-term visual outcome between IVA+PDT group and IVR+PDT group; however, there were no significant differences in visual outcomes between the 2 groups."

7. Are there any cases presenting with massive subretinal hemorrhage with or without vitreous hemorrhage? Do you still use combination therapy at the first place, instead of use of intravitreal tPA or gas? Or you exclude this kind of cases? If so, you should add exclusion criteria.

8. If the case with dense cataract and PCV as the first presentation, how did you do? Or you exclude this kind of cases?

9. When you use intravitreal anti-VEGF, do you exclude the cases with recent thromboembolic events or pregnancy?

Reply to the question 7-9: We added new sentences in the Materials and Methods section as follows " Exclusion criteria was the case with massive subretinal hemorrhage with or without vitreous hemorrhage, dense cataract, history of recent thromboembolic events, or pregnancy."

10. Table 1 did not have the range of Age, Baseline log MAR BCVA, Greatest linear dimension (μm), Central Retinal Thickness (μm), and Subfoveal Choroidal Thickness

Reply: Thank you for the advice. we revised Table 1 adding the range of Age, Baseline log MAR BCVA, GLD, CRT, and SCT.

11. Can you list in a table or describe in the article about final log MAR BCVA, Greatest linear dimension (μm), Central Retinal Thickness (μm), and Subfoveal Choroidal Thickness

Reply: We added new sentences to point out the final BCVA in the Results section as follows " Mean log MAR BCVA improved from 0.55±0.28 at baseline to 0.40±0.40 at 60-month." We also added new sentences in the Results section to point out the final CRT as follows "Fig 4 shows changes of central macular thickness after the initial combination therapy. Compared with baseline value, central macular thickness significantly decreased (p=<0.001) throughout the follow-up period and mean CRT decreased from 368±100 µm at baseline to 217±94 µm at 60-month." The final mean SCT was shown in Table 8.

Reviewer #2: It is useful to publish the data presented as the long term (5 year) outcomes are valuable information. The combination of anti-VEGF + PDT is at present a treatment of choice for PCV. I have the following remarks:

Replay: Thank you for your valuable comments and a positive feedback.

1. Line 44: Please leave out the statement starting with "To date PCV has been considered ...". This is not generally true.

Reply: Thank you for the advice. We deleted these sentences.

2. In the description of the PDT important parameters are left out. Over what time was the injection performed? How long was the delay between injection and light application?

Reply: We added new sentences in the Materials and Methods section to describe the combination therapy as follows " All participants received intravitreal injection of ranibizumab (0.05mg/0.05ml) or aflibercept (0.2mg/0.05ml) 1 week before PDT(1 injection and 1 PDT)."

3. At the bottom of page 5: Please describe more precisely how the follow-up treatments were done. When was only anti-VEGF applied, at what frequency, and what conditions? When combination therapy was applied in the follow-up, did anti-VEGF precede PDT, etc.

Reply: Thank you for the comments. We described the follow-up treatments in detail in the Materials and Methods section as follows "Follow-up examination included assessment of BCVA using Landolt chart, intraocular pressure, biomicroscopy with or without a 76 D lens, and SD-OCT, and was performed every 3 months until recurrent exudation developed. Recurrence was defined as newly developed hemorrhage on fundoscopy or subretinal fluid detected by SD-OCT. Additional FA/ICGA was performed when recurrent exudation was seen. When ICGA showed residual or recurrent polypoidal lesion, additional combination therapy (1 injection and 1 PDT) was administrated in the same fashion as the initial treatment. When ICGA exhibited abnormal vascular network without polyp, additional intravitreal injection of anti-VEGF agent was administrated. After first recurrence, patients were followed every month and PRN treatment was performed until exudation including subretinal fluid and intraretinal fluid was completely absorbed."

4. There are very important differences in the outcome when compared with the data of reference 13 (Miyata et. al.). This needs to be discussed in some detail.

Reply: Thank you for the valuable comments. We added new sentences in the Discussion section as follows "Miyata et al. reported 5-year results after combination therapy involving PDT and intravitreal injection of ranibizumab for 20 eyes with PCV. In their report, BCVA significantly improved at 1 year, but thereafter BCVA declined and returned to baseline value at 5 year. Similarly, in the present study, there has been the trend of BCVA deterioration after 3 years from initial treatment, but the significant BCVA improvement was maintained through 5 years. One possible reason for the differences of visual outcome between the 2 studies is that, in this study, mean age of the patients was younger than that in the Miyata's report. We reported that in the previous study, older age was associated with recurrence of PCV lesions after the combination therapy. The difference of mean age between the 2 studies might affect the visual outcome. Differences in baseline characteristic between the 2 studies might result in different results.[18] "

5. The literature reports are rather incomplete. The literature reported is to a significant extent self-citation. See for instance P. Jain et. al. Indian J. Ophthalmol. 2018, Jingyuan Yang et. al. BMC Ophthalmology December 2019, and Qian T. et. al. Eur. J. Clin. Invest. 2018. among others.

Reply: Thank you for the advice. On the basis of your suggestions, we added the articles suggested above. Line 272-274,” Several studies reported long-term results of the combination therapy involving PDT and anti-VEGF agents for PCV. [13,16,17]”

6. Discussion of the influence of the ARMS2 gene as done by the present authors is of interest. Why not also discuss the results in terms of the subtypes PCV1 AND PCV2.

Reply: Thank you for the suggestion. To reply this point, we added new sentences in the Discussion section as follows " Thirdly, early ICGA images were not performed for all patients, therefore we could not judge presence or absence of feeder vessels of polypoidal lesions. For this reason, we did not distinguish the PCV subtypes. A large-scale prospective study would be needed to confirm the present tentative conclusion."

7. Other authors have done this combination therapy by treating first with PDT and the following up with anti-VEGF. Please discuss your choice, and why you think it is preferable.

Reply: Thank you for the advice. In order to discuss our choice about the initial combination therapy, we added new sentences in the Discussion section as follows " In the present study, intravitreal injection of anti-VEGF agents was administrated a week before PDT. The reason why we chose this method was that, PDT causes photochemical thrombotic occlusion of polypoidal lesions, while also induces the upregulation of VEGF as an adverse side effect. Initial injection of anti-VEGF agents is expected to decrease the high intraocular concentration of VEGF, which causes exudation of both intraretinal and subretinal fluid."

---

## [Decision Letter · Decision Letter 1]

3 Feb 2020

Five-year outcomes of photodynamic therapy combined with intravitreal injection of ranibizumab or aflibercept for polypoidal choroidal vasculopathy

PONE-D-19-34381R1

Dear Dr. Sakurada,

We are pleased to inform you that your manuscript has been judged scientifically suitable for publication and will be formally accepted for publication once it complies with all outstanding technical requirements.

With kind regards,

Michael R Hamblin

Academic Editor

PLOS ONE

Additional Editor Comments (optional):

Reviewers' comments:

Reviewer's Responses to Questions

**Comments to the Author**

1. If the authors have adequately addressed your comments raised in a previous round of review and you feel that this manuscript is now acceptable for publication, you may indicate that here to bypass the “Comments to the Author” section, enter your conflict of interest statement in the “Confidential to Editor” section, and submit your "Accept" recommendation.

Reviewer #1: All comments have been addressed

Reviewer #2: All comments have been addressed

2. Is the manuscript technically sound, and do the data support the conclusions?

Reviewer #1: Yes

Reviewer #2: (No Response)

3. Has the statistical analysis been performed appropriately and rigorously? 

Reviewer #1: Yes

Reviewer #2: (No Response)

4. Have the authors made all data underlying the findings in their manuscript fully available?

Reviewer #1: Yes

Reviewer #2: (No Response)

5. Is the manuscript presented in an intelligible fashion and written in standard English?

Reviewer #1: Yes

Reviewer #2: (No Response)

6. Review Comments to the Author

Reviewer #1: The questions were properly answered. The paper can be accepted. The outcome demonstrated combined PDT and IVA can be long-term as effective as PDT and IVR.

Reviewer #2: (No Response)

7. PLOS authors have the option to publish the peer review history of their article (what does this mean?). If published, this will include your full peer review and any attached files.

Reviewer #1: No

Reviewer #2: No

---

## [Editor Report · Acceptance letter]

5 Feb 2020

PONE-D-19-34381R1 

Five-year outcomes of photodynamic therapy combined with intravitreal injection of ranibizumab or aflibercept for polypoidal choroidal vasculopathy 

Dear Dr. Sakurada:

I am pleased to inform you that your manuscript has been deemed suitable for publication in PLOS ONE. Congratulations! Your manuscript is now with our production department. 

With kind regards,

on behalf of

Dr. Michael R Hamblin 

Academic Editor

PLOS ONE